# Noninvasive investigation of the cardiodynamic response to 6MWT in people after stroke using impedance cardiography

Fang Liu[1☉], Alice Y. M. Jones[2,3‡*], Raymond C. C. Tsang[4‡☉], Yao Wang[1,5☉], Jing Zhou[1☉], Mingchao Zhou[1☉], Yulong Wang[1☉]

**1** Department of Rehabilitation, Health Science Center,Shenzhen Second People's Hospital, The First Affiliated Hospital of Shenzhen University, Shenzhen, China, **2** School of Health and Rehabilitation Sciences, The University of Queensland, Brisbane, QLD, Australia, **3** Faculty of Health Sciences, The University of Sydney, Sydney, NSW, Australia, **4** Department of Physiotherapy, MacLehose Medical Rehabilitation Centre, Hong Kong, Hong Kong, **5** Department of Rehabilitation, Shenzhen Dapeng New District Nan'ao People's Hospital, Shenzhen, China

☉ These authors contributed equally to this work.
‡ These authors also contributed equally to this work.
* alice.jones@sydney.edu.au

**Data Availability Statement:** All relevant data are within the paper and its Supporting Information files.

## Abstract

This is a cross-section observational study that investigated the cardiodynamic response to a 6-minute walk test (6MWT) in patients after stroke using impedance cardiography (ICG). Patients diagnosed with stroke were invited to participate in a 6MWT on consecutive days. Heart rate (HR), cardiac output (CO), stroke volume (SV) and cardiac index (CI) were measured by ICG using the PhysioFlow® PF07 Enduro™ at 1-second intervals for 10 minutes prior to, during and for 10 minutes after each 6MWT. Oxygen saturation, perceived exertion score (modified Borg scale) and the distance covered at the end of each 6MWT were recorded. Twenty-nine patients (mean age 55.6±10.9 years) completed the study. The mean duration of stroke after diagnosis was 14.4±19.1 months. There were no differences in the measured data between the first and second 6MWT (mean intraclass correlation coefficient (ICC) range: 0.87–0.95). The 6 minute walk distance (6WMD) covered in the two 6MWTs was 246±126 and 255±130m respectively (p>0.05). Mean measured data for each subject at rest, and at the end of the better performed 6MWT were, respectively: HR 78±11 and 100±18 bpm; CO 5.5±1.2 and 8.9±2.6 l/min, SV 71.3±16 and 89.3±18.6 ml/beat and CI 3.0±0.6 and 4.9±1.3 l/min/m². After commencement of the 6MWT, the increase in SV took 30 sec before the rise approaching a plateau, whereas HR, CO and CI continued to rise steeply for 90 sec before leveling off to a steady rise. After completion of the 6MWT, all parameters had returned to baseline by a mean of 3.5 min. Sub-group analysis showed that the increase in cardiac output was predominantly contributed by an increase in heart rate in participants diagnosed with stroke for less than 1 year, whereas both stroke volume and heart rate contributed similarly to the increase in cardiac output in participants with diagnosis of stroke for longer than 1 year. Systolic blood pressure (SBP) and diastolic blood pressure (DBP) both returned to baseline within 2 minutes post 6MWT. HR recorded at the end of the 6MWT was 60.8±10.6% of the predicted maximal heart rate and perceived exertion score

**Funding:** This study was funded by a grant from Sanming Project of Medicine in Shenzhen awarded to YM (No. SZSM201512011).

**Competing interests:** No authors have competing interests.

was 5±2. Correlations between 6MWD and HR, and between 6MWD and SV were weak, with correlation coefficients Spearman's rho ($r_s$) =0.46, and 0.42, respectively ($p<0.05$). Correlation between 6MWD and CO and CI were higher ($r_s$= 0.66 and 0.63, respectively ($p<0.01$)). This is the first study to report cardiac responses during a 6MWT in stroke patients. ICG is a reliable, non-invasive, repeatable method of measuring cardiodynamic data in stroke patients.

## Introduction

Neurological deficits associated with stroke often lead to significant physical disability [1], resulting in low ambulatory activity and poor cardiovascular fitness [2]. Low cardiovascular fitness in stroke survivors may not only lead to low social participation, effecting quality of life [3], but may also worsen underlying cardiovascular and metabolic risk factors such as hypertension, obesity, diabetes and dyslipidemia, resulting in an increased risk of recurrent stroke [4]. To counteract the decline in aerobic fitness in this population, rehabilitation programs to improve aerobic capacity in people post stroke have been recommended [5,6]. Appropriate prescription of an effective exercise program requires accurate monitoring and evaluation of aerobic capacity. Assessment of peak oxygen uptake ($VO_2$peak) by indirect calorimetry during progressive cardiopulmonary exercise testing (CPET) is the gold standard for evaluation of aerobic capacity [7], however this is invariably impractical in patients with stroke. Stroke-specific impairments such as muscle weakness, fatigue, poor balance, contracture and spasticity often limit the patient reaching their maximum capacity using these standard exercise tests. The mean $VO_{2peak}$ measured during a 6MWT is similar to cycle graded exercise test value [8]. In clinical practice, the 6-minute walk test (6MWT) is therefore often used as a sub-maximal test to assess walking capacity and cardiovascular fitness in this group of patients [9–11]. Whether 6MWT is an adequate measure of aerobic fitness has been queried [12], but there is strong evidence to support the reliability and construct validity of using the 6MWT in people after stroke [13,14]. Heart rate was reportedly 85% of maximum at the end of a 6MWT, while oxygen consumption was 70% of peak values, after a progressive, standardized exercise test, in patients with stroke [15, 16]. These studies suggest that the 6MWT induces a significant aerobic challenge in a stroke patient.

While the 6MWT reflects the aerobic capacity of a patient after stroke, the cardiodynamic parameters (stroke volume, cardiac output and cardiac index) which determine exercise capacity during a 6MWT, have not been reported in people after stroke. Impedance cardiography (ICG) is an established noninvasive technique used to measure various indices of cardiovascular function and correlates closely with invasive techniques using dye dilution [17]. The validation and recent advances in clinical applications of ICG are well described [18,19], although the role of ICG was not shown to be superior to echocardiography [20]. ICG measurements during a 6MWT in people after stroke however have not been reported.

The objectives of this study in people with stroke were to investigate: (1) the cardiodynamic responses prior, during and after 6MWT using ICG; and (2) the relationship between the 6-minute walk distance (6MWD) and the cardiodynamic parameters during the 6MWT. Lastly, the variability of ICG data measured between two 6MWTs was also examined.

## Material and methods

Approval to conduct this study was obtained from the Institutional Review Board of Shenzhen Second People's Hospital (Ethics approval number: 20190605016-FS20190629008). The protocol for this study is available at dx.doi.org/10.17504/protocols.io.6s2hege

## Study design

This study adopts a cross-section observational design involving patients diagnosed with stroke and received regular medical follow ups from July 1st to August 31st 2019, at the Rehabilitation Department at the Shenzhen Second People's Hospital, China. Patients were invited to undertake two 6MWTs in consecutive days at the hospital, during which cardiodynamic parameters were measured by ICG.

**Inclusion and exclusion criteria.** The inclusion criteria were: (1) age $\geq$ 18 years, (2) clinically diagnosed with ischemic and/or hemorrhagic stroke, (3) time lapsed after stroke diagnosis $\geq$ 1 month, (4) patient able to independently ambulate with or without an assistive device for $\geq$ 100 meters, (5) medically stable and with no significant pain limitations, (6) able to clearly comprehend exercise testing instructions.

The exclusion criteria were: (1) patients prescribed with regular beta blockers or those requiring beta blocker at the time of the study; (2) other neurological or orthopaedic conditions that may cause motor deficit (e.g. fracture, degenerative joint changes, or clinical instability of the hip or knee joint), (3) psychiatric impairment, such as severe depression or panic disorder, (4) pregnancy, (5) uncontrolled hypertension, arrhythmia, or an unstable cardiovascular status as advised by the attending physician.

**Sample size.** Sample size was estimated by Pass 11 (NCSS, LLC, Kaysville, Utah). To achieve a power of 0.8 and intraclass correlation coefficient value > 0.5 for reliability, the minimum estimated number of participants was 22. Allowing for a possible withdrawal rate of 20%, a minimum of 27 participants was deemed necessary [21].

## Procedure

The aims and procedures of the study were explained to the participants and written informed consent obtained. Demographic data for each participant including age, gender, height, weight, body mass index (BMI) and lean body mass were recorded. The stroke diagnosis, time elapsed after stroke diagnosis, past medical history (e.g. hypertension, diabetes mellitus, cardiovascular disease, lipidemia, kidney disease, pulmonary disease), the National Institute of Health Stroke Scale (NIHSS) and Modified Rivermead Mobility Index (MRMI) were retrieved from each patient's medical record [22, 23].

Participants were invited to perform a 6MWT at the cardiopulmonary laboratory of the hospital on two consecutive days at a time > 2 hours after a light meal. Participants were requested to avoid caffeine-containing products, nicotine, and alcohol for at least 12 hours before attending the laboratory. The patients performed the 6MWT in a 30m indoor hallway located immediately outside the laboratory. The 6MWT was conducted according to the standard protocol recommended by the American Thoracic Society (ATS) [11]. Prior to the 6MWT, each participant was asked to rest in a sitting position for 10 minutes during which hemodynamic parameters (ICG, see below), oxygen saturation ($SpO_2$) (Heal Force pulse oximeter, POD-3, China), blood pressure (BP) (OMRON electronic blood pressure monitor, U30, China), perceived fatigue sensation (modified Borg 0–10 Scale [24]), were recorded. Blood pressure was measured immediately before and after the 6MWT, and at 2-minute intervals during the 10-minute rest period after the 6MWT. $SpO_2$ was measured immediately before, at each minute during, and at the end of the 6MWT. Perceived exertion level was recorded immediately before and at the end of 6MWT.

**Measurement of cardiac parameters.** Heart rate (HR), stroke volume (SV), cardiac output (CO) and cardiac index (CI) were measured by Impedance cardiography using the PhysioFlow®PF07 Enduro™ (PhysioFlow Enduro, Paris, France). Auto-calibration of the machine was performed as instructed by the manufacturer, prior to data collection. The Enduro™ is a

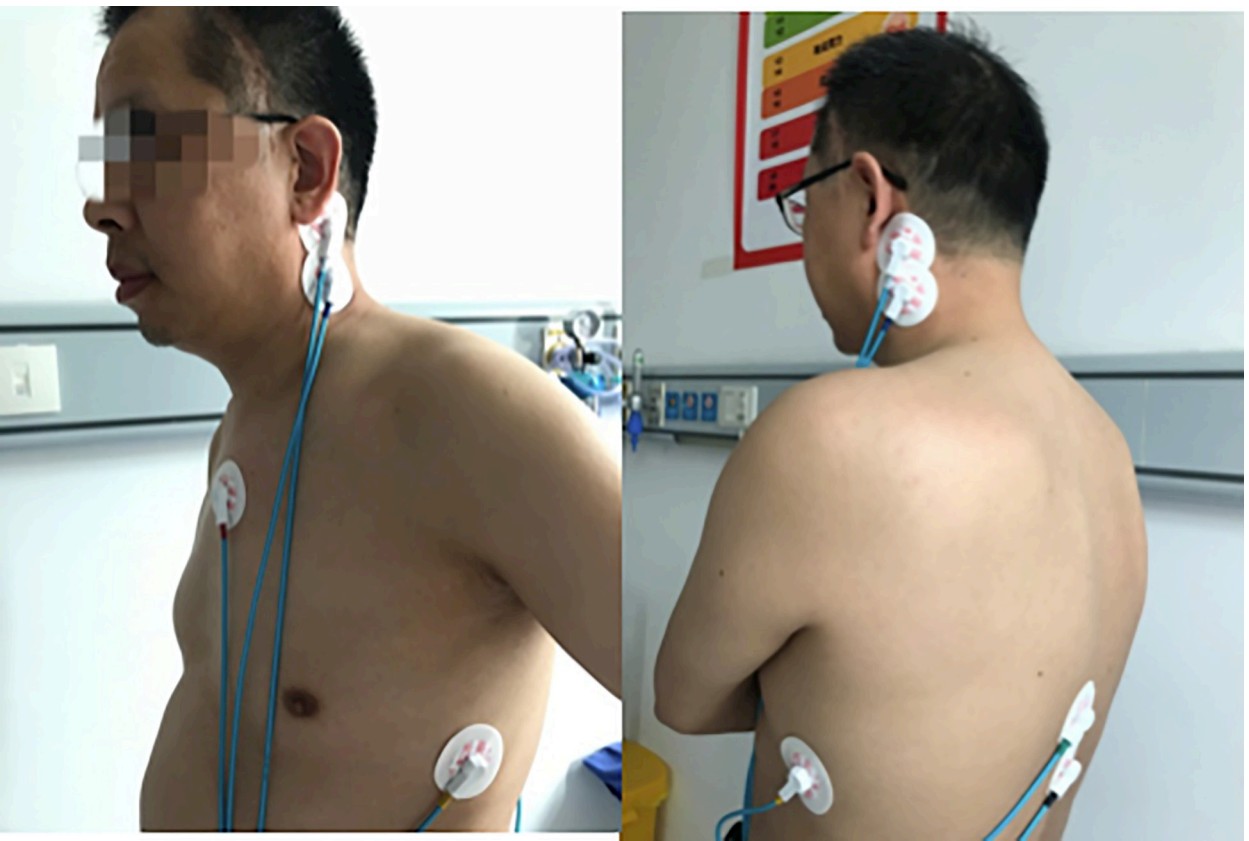

**Fig 1. Placement of electrodes (with patient consent).**

portable, non-invasive device that adopts real-time wireless monitoring of morphology-based impedance cardiography signals via a blue tooth USB adapter. Variations in the impedance signal during cardiac ejection generates a specific waveform from which the SV is calculated when an alternating high-frequency, low magnitude current is passed across the thorax [25]. CO (l/min) is calculated by multiplication of the SV and HR, cardiac index (CI) is computed by dividing the CO by calculated body surface area based on height and weight of the patient; heart rate, and R-R interval were derived from the electrocardiograph (ECG) [26]. HR, SV, CO, CI data were computed by the Enduro™ at 1-second intervals from 10 min prior, during, and for 10 min after the 6MWT.

**Electrode placement.**   Electrode placement was conducted as described by Tonelli and colleagues [27] (Fig 1).

## Data analyses

HR, SV, CO, CI were averaged every 10 seconds. The data at rest, at every $30^{th}$ sec during the 6MWT, and for 10 minutes during recovery, were collected for analysis. All data were analyzed using IBM SPSS Statistics for Windows, Version 23.0 (Armonk, NY: IBM Corp). Demographic data and clinical characteristics for all participants were summarized using descriptive statistics. Wilcoxon signed rank test was used to compare the differences between the first and the second 6MWT for $SpO_2$, Borg score and 6MWD data. Changes in variables, the time taken for each parameter to reach a steady plateau and the time taken to return to baseline, were analyzed using repeated-measures ANOVA with post-hoc Least Significant Difference (LSD)

analysis. The intraclass correlation coefficient (ICC$_{1,1}$) [28] for parameters recorded between the first and second 6MWT was calcuated by a one-way random-effects model. Standard error of measurement (SEM) was calculated by taking the square root of the mean square error of repeated measurements ($\sqrt{MSE}$) [29]. Correlation analysis was conducted using Spearman's rho (r$_s$) to determine the relationship between 6MWD and cardiac variables at the end of the 6MWT. The predicted maximal heart rate was estimated by applying the formula, 220 minus age. The time elapsed since their stroke event in our participant cohort ranged from 1 month to >60 months. Subgroup analysis with linear regression was performed in participants whose elapsed time since stroke was ≤12 months or >12 months, using CO as the dependent variable and HR change and SV change as covariates to determine whether HR or SV was the main determinant factor for changes in CO.

All patient records were locked in special storage by the chief investigator LF. Data entered for analysis were de-identified.

## Results

Thirty-one subjects were recruited for the study. Two subjects were unable to return for a second 6MWT as they had been transferred to another hospital. A total of 29 patients (mean age 55.6±10.9 years) completed both 6MWTs. Patient demographics and clinical characteristics are displayed in Table 1. There was no statistical difference in recorded haemodynamic data between the first and second 6MWT; intraclass correlation coefficient (ICC) range was 0.87–0.95 (S1 Table). The mean distance covered in the second 6MWTs was slightly higher

**Table 1. Demographic data of the 29 participants.** Data in n(%) or mean±SD.

| Gender | | Male = 21 (72.4%) |
|---|---|---|
| | | Female = 8 (27.6%) |
| Age (years) | | 55.6±10.9 |
| Height (cm) | | 168.4±8.1 |
| Weight (kg) | | 70.2±9.8 |
| Lean body mass (kg) | | 52.3±7.1 |
| Diagnosis | | |
| | Cerebral hemorrhage | 11 (37.9%) |
| | Cerebral infarction | 18 (62.1%) |
| Duration of stroke (months) | | |
| | mean | 14.4±19.1 |
| | 1–3 | 9 (31.1%) |
| | 4–12 | 10 (34.5%) |
| | 13–24 | 5 (17.2%) |
| | 48–60 | 5 (17.2%) |
| With hypertension | | 23 (79.3%) |
| With diabetes | | 6 (20.7%) |
| With hyperlipidemia | | 12 (41.4%) |
| Ambulation status | | |
| | Free | 24 (82.8%) |
| | With a cane | 5 (17.2%) |
| NHISS score | | 4.3±2.9 |
| MRMI score | | 36.7±3.5 |

NHISS = National Institutes of Health Stroke Scale; MRMI= Modified Rivermead Mobility Index

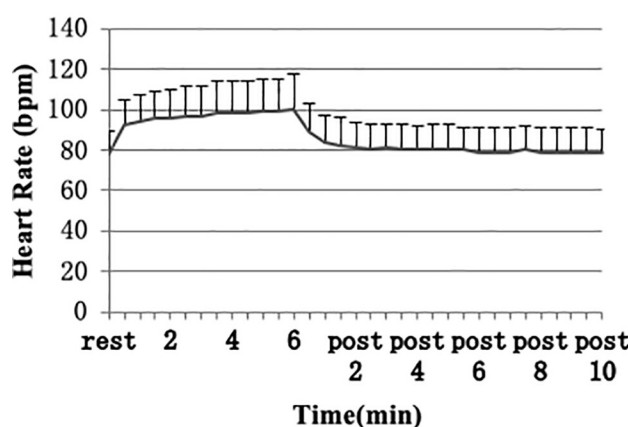

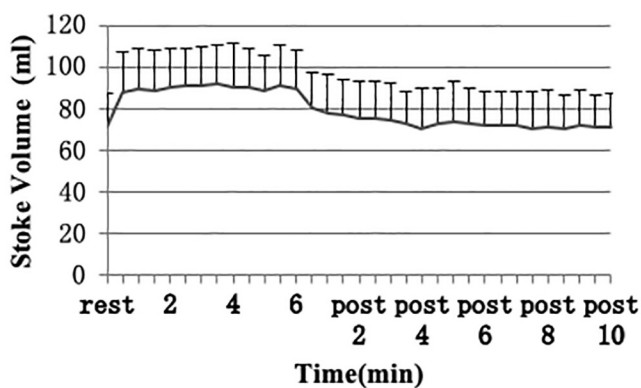

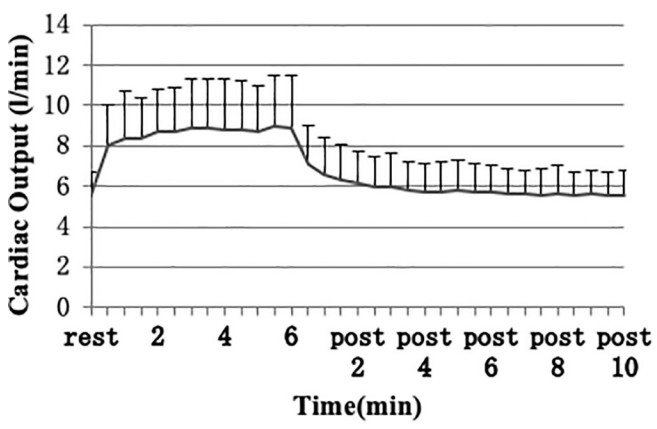

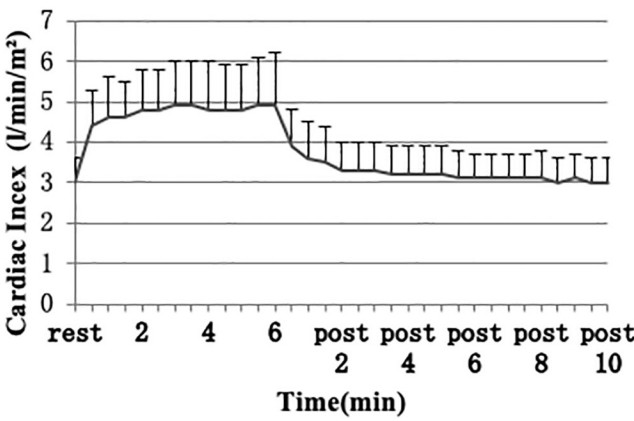

**Fig 2. Cardiodynamic changes at rest, during and post 6MWT.**

compared to the first test, but this did not reach a statistical significance [mean distance covered in the two tests was 246±126 and 255±130m, respectively (p>0.05)]. For each subject, the data recorded in the 'better' walk test were used to describe the cardiodynamic response to a 6MWT. SV rose quickly during the first 30 sec, while HR, CO and CI continued to rise sharply till 90 sec before the rise became steadier and approaching a plateau; all parameters had returned to baseline by a mean of 3.5 min post 6MWT (S2 Table). Graphical presentations of the changes in parameters are displayed in Fig 2. Mean cardiodynamic data are illustrated in S3 Table.

Oxygen saturation remained stable throughout the 6-minute walk. Blood pressures immediately before and after 6MWT are displayed in S2 Table. Rate of exertion at the end of the 6MWT expressed as a modified Borg scale, was 5±2 on both days. HR recorded at the end of the 6MWT was 60.8±10.7% of the predicted maximal heart rate. Correlations between 6MWD

**Table 2. Heart rate and cardiodynamic parameters during a 6MWT reported in previous literature in comparison to data recorded in the current study.**

| Reference | current study | Muren et al [32] | Pradon et al [31] | Kubo et al [22] | Salbach et al [5] | Harmsen et al [30] | Someya et al [33] | | Tonelli et al [27] | |
|---|---|---|---|---|---|---|---|---|---|---|
| Age | 55.6±10.9 | 58±9 | 53.4±13.7 | 72±10.7 | 71.1±9.7 | 53.0±8.9 | 20.5±0.7 | 60.2±6.1 | 51.8 ±15 | 49 ± 11 |
| Pathology | Stroke | Stroke | Stroke | Stroke | Stroke | ASAH | Young adults | Elderly | Healthy | PH |
| Mean Duration of stroke | 14.4±19.1 (mon) | 60 ±27 (mon) | 16±8 (mon) | 5.1 ± 2.6 (days) | 2.0 ± 1.1 (years) | - | - | - | - | - |
| 6MWD (m) | 255±130 | 353 | 273.8 | 331.8 | 254.9 | 498 | 541 | 533 | 560 | 380 |
| HR-rest (bpm) | 78±11 | 80±12 | 72.3±11.1 | 74.6 ±12.2 | 76.1 ± 10.1 | - | 75.8±12.0 | 77.3 ±11.6 | 76±14 | 83±12 |
| HR-end of 6MWT | 100±18 | 105±19 | 106.21±21.41 | 82.1±14.3 | 103.7 ± 13.0 | 114±20 | 117.4±26.7 | 121.0 ±21.8 | 127±18 | 117 ±20 |
| HR change (%) | 28.7±22.1 | 31.25 | 46.92 | 10.05 | 36.27 | - | 54.88 | 56.53 | 67.11 | 40.96 |
| SV-rest (ml) | 71.3±16 | - | - | - | - | - | 89.4±22.1 | 68.9 ±11.6 | 62±19 | 51±17 |
| SV-end of 6MWT (ml) | 89.3±18.6 | - | - | - | - | - | 111.3±27.3 | 102.9 ±15.2 | 95±26 | 75±22 |
| SV change (%) | 26.1±14.2 | - | - | - | - | - | 24.50 | 49.35 | 53.23 | 47.06 |
| CO-rest (l/min) | 5.5±1.2 | - | - | - | - | - | 6.6±1.3 | 5.3±1.1 | - | - |
| CO-end of 6MWT (l/ min) | 8.9±2.6 | - | - | - | - | - | 12.7±2.9 | 12.5±2.9 | - | - |
| CO change (%) | 64.1±40.5 | - | - | - | - | - | 92.42 | 135.85 | - | - |
| CI-rest (l/min/m$^2$) | 3.0±0.6 | - | - | - | - | - | 4.0±0.6 | 3.4±0.6 | 2.3±0.9 | 2.3±0.7 |
| CI-end of 6MWT (l/ min/m$^2$) | 4.9±1.3 | - | - | - | - | - | 7.6±1.4 | 7.9±1.7 | 5.9±1.7 | 4.9±1.5 |
| CI change (%) | 63.3±41.1 | - | - | - | - | - | 90 | 132.35 | 156.52 | 113.04 |
| %MHR at end of 6MWT | 60.8±10.6 | - | - | - | - | 67% | 59% | 76% | - | - |
| Borg scale at end of 6MWT | 5±2/10 | - | 11.4/20 | 2.2/10 | 3.1/10 | - | - | - | 1.2/10 | 2.6/10 |

6MWD=6 minute walk distance; 6MWT=6 minute walk test; HR=heart rate; SV=stroke volume; CO=cardiac output; CI=cardiac index; mon=months; numbers in brackets are reference number; ASAH= aneurysmal subarachnoid hemorrhage; PH=pulmonary hypertension; %MHR= percentage of predicted maximal heart rate.

and HR, and between 6MWD and SV were weak, (correlation coefficients $r_s$=0.46, and 0.42, respectively ($p<0.05$)). Correlation between 6MWD and CO, and with CI, were higher ($r_s$=0.66 and 0.63, respectively) ($p<0.01$) (S4 Table). Table 2 illustrates previous reported cardiodynamic parameters measured during a 6MWT [5, 22, 27, 30–33] compared to data reported in the current study.

Subgroup analysis showed that if the time elapsed since the stroke event was 12 months or less, the change in HR contributed 60% to the increase in CO with 6MWT exercise, while the unique SV contribution to the increase in CO, was only 12%. In the 10 participants in whom > 12 months had elapsed since their stroke, HR and SV increase contributed equally to CO increase with exercise (Table 3). Although the BSA derived from height and weight data differed, there was no significant between-group difference in age, BMI, haemodynamic variables at rest or at the end of the 6MWT (S5 Table). Age and gender were initially included as independent variables in the regression analysis, however they were eliminated from the regression model due to statistical non-significant effect. Fig 3 illustrates the cardiodynamic responses to the 6MWT in the two groups.

**Table 3. Linear regression using cardiac output change as dependent variable and HR change or SV change as covariates.**

| | Time elapsed after stroke less than 1 year (n=19) | Time elapsed after stroke over 1 year (n=10) | Total (n=29) |
|---|---|---|---|
| HR change standardized beta coefficient | 0.821 | 0.559 | 0.695 |
| SV change standardized beta coefficient | 0.361 | 0.539 | 0.474 |
| HR change semipartial correlation | 0.777 | 0.425 | 0.626 |
| SV change semipartial correlation | 0.342 | 0.410 | 0.427 |
| HR change unique contribution to CO change (%) | 60 | 18 | 39 |
| SV change unique contribution to CO change (%) | 12 | 17 | 18 |
| 6MWD (m) | 265±136.3 | 222±121.5 | 255±130 |

6MWD=6-minute walk distance; 6MWT=6-minute walk test; HR=heart rate; SV=stroke volume; CO=cardiac output; CI=cardiac index

## Discussion

There are studies reporting the heart rate changes during a 6MWT in people after stroke [22,31,32], but this is the first study to report changes in stroke volume, cardiac output, cardiac index, together with heart rate, in a stroke population during a 6MWT. The mean distance covered by our cohort of patients was 255m; which was slightly less, but comparable to 274m reported by Pradon and colleagues in a cohort of stroke patients of similar age [31]. While the 6MWDs were similar, the increase in heart rate at the end of the 6MWT in our patient cohort was 29%, compared to 47% reported in Pradon's study (Table 2). The subjective rate of exertion was 5/10 (modified Borg scale) in our cohort and 11/20 (Borg scale) in Pradon's study. It would appear that Pradon's cohort achieved similar distance to our cohort with a higher increase in heart rate but lower rate of perceived exertion. The perceived exertion level reported by our subjects was equivalent to an intensity level described as 'hard', although the peak HR generated was only 60% of the subject's predicted maximal heart rate. All our participants were able to mobilise independently with or without a walking stick, although some were unable to independently negotiate stair climbing. The mean MRMI score of our participants was 36.7; very close to the full scale of 40, reflecting the high level of independence during transfers and walking. With no cardiodynamic data available from Pradon's study for comparison, we are unable to explain the differences between ours and Pradon's study, but it may be worthy to note that, unlike in western countries, stroke rehabilitation programs for patients in China are relatively *undemanding* and mainly involved leisure walking and passive limb exercise. This may explain the higher perceived exertion score in our patients during the 6MWT; as they were not accustomed to such a strenuous challenge. Muren and colleagues [32] report a lower percentage (31%) increase in heart rate, similar to our cohort, but Muren's subjects were slightly older (58 years) and with a longer elapsed time since their stroke event.

Application of ICG measurement during a 6MWT allows a more informed analysis of the correlation between the 6MWD and HR, SV, CO and CI. Our study showed that in a population of people after stroke, the 6MWD demonstrated a positive correlation with HR, SV, CO and CI (S4 Table). The relationship between 6MWD and HR and CI, reported in subjects with pulmonary hypertension by Tonelli and colleagues [27], was very similar to ours. However, the coefficient values for SV and CO were not provided by these authors. The correlation

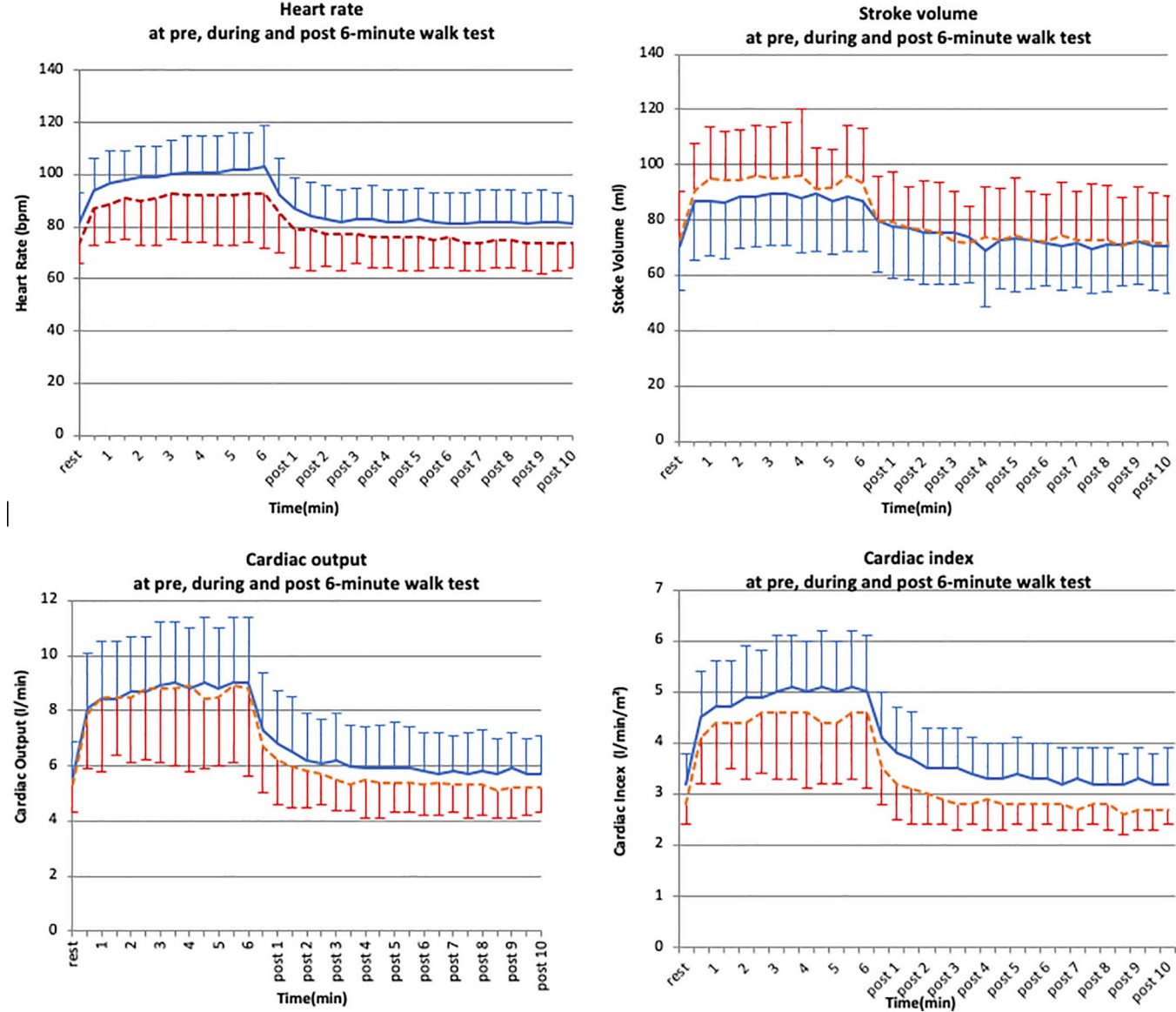

**Fig 3. Cardiodynamic changes during the 6MWT in participants post-stroke duration of less or longer than 1 year.** Solid line= time elapsed after stroke less than 1 year; dotted line=time elapsed after stroke longer than 1 year.

between 6MWD and HR and CO in healthy adults reported by Someya and colleagues was comparable to our findings [33], but their study showed that the 6MWD was not correlated with stroke volume. We posit that this may be because the correlation data in Someya's subjects were pooled from both young and elderly subjects, and that the SV in their young subjects did not change during the 6MWT [33].

Someya and colleagues compared changes in haemodynamic variables between young and elderly healthy subjects after a 6MWT [33]. Their data demonstrated that the increase in HR took more than 60s in the elderly, but took less than 30s to reach a plateau in the young, the authors explained this being the reason of the slower response to changes in cardiac output in their elderly subjects. Data from our stroke subjects showed that HR took 90sec to rise before reaching a more steady state, while the increase in SV steadied after only 30 seconds into the

6MWT. As the patterns of increase in HR and CO were similar (S2 Table), this suggests that the increase in CO during the 6MWT was met by an increase in HR rather than SV in our subject cohort. This proposition is supported by results of linear regression analysis as reported by Tonelli et al [27]. Although Tonelli et al [27] and Somya et al [32] do not report data obtained from people after stroke, their work explains the analysis of cardiodynamic responses acquired using ICG during a 6MWT.

The range of time elapsed since the stroke event was large in our cohort. While the changes in CO were similar in the two sub-cohorts during the 6MWT, analysis of our data showed that the unique contribution to CO by HR was much higher (60%) in the group where the onset of their stroke event occurred < 12 months, compared to the 18% contribution in those subjects where the elapsed time after stroke diagnosis longer than 1 year (Table 3). In the group with time elapsed after stroke > 12 months, SV changes in contribution to CO were greater than those with shorter time elapsed after stroke diagnosis, although the difference is less dramatic (17% vs 12%) (Table 3). A lower contribution of SV to haemodynamic response may be associated with poorer diastolic function [34], and left ventricular diastolic dysfunction associated with poor functional outcomes and vascular events has been reported in a stroke population [35]. Exercise training improves ventricular systolic and diastolic function [36], thus we speculate that the diastolic function of those persons in our patient cohort with a longer elapsed time after stroke may have improved in accord with the longer exposure to exercise training, thereby contributing to the increase in CO during the 6MWT. This might further endorse the role of ICG in investigation of the effect of rehabilitation in people after stroke.

Despite a slight increase in the distance covered over the second 6MWT, there was no statistically significant difference in the distance covered. This is not surprising because the reliability of a 6MWT is high and hence a single 6MWT is recommended in people after stroke [14]. Notwithstanding a small 'practice effect' of the two 6MWT, the high ICC values of ICG data recorded on consecutive days provide support for the reliability of ICG measurement of cardiodynamic variables.

One limitation of our study is the small sample size and that we have a large range of time elapsed since the stroke event in our subject cohort. The number of subjects with onset of stroke duration >12 months was only 10. While our data suggest the increase in CO was predominantly a consequence of HR increase in subjects where the time elapsed after stroke was less than 1 year, the data failed to reach statistical significance, possibly due to the small sample size. Linear regression analysis showed no effect of age and gender on our measured cardiodynamic variables, again probably due to the small sample size. Age and gender can influence cardiovascular function [37,38], and further studies with a larger sample size are required to explore any effect of age and gender on cardiodynamic parameters, measured by ICG, in response to exercise training, in patients recovering from stroke.

## Conclusion

This is the first study to report changes in cardiodynamic parameters during a 6MWT in a cohort of stroke patients using impedance cardiography. We demonstrated that 6MWD correlated with the increase in CO and that this increase was in response to a change in HR rather than SV in participants with diagnosis of stroke less than 1 year, however in participants with diagnosis of stroke greater than 1 year, stroke volume and heart rate both contributed similarly to the increase in cardiac output. Findings of this current study suggests further investigation of cardiodynamic response to exercise in stroke patients by impedance cardiography are warranted.

## Supporting information

**S1 Checklist. STROBE statement—checklist of items that should be included in reports of observational studies.**
(DOCX)

**S1 Table. ICC data.**
(DOCX)

**S2 Table. Time for each variable returned to baseline.**
(DOCX)

**S3 Table. Mean cardiodynamic data.**
(DOCX)

**S4 Table. Correlation between 6MWD and cardiodynamic parameters at the end of 6MWT.**
(DOCX)

**S5 Table. Comparison of characteristics in subjects with time elapsed since stroke diagnosis longer or shorter than 1 year.**
(DOCX)

## Author Contributions

**Conceptualization:** Fang Liu, Alice Y. M. Jones, Yulong Wang.

**Data curation:** Fang Liu, Yao Wang, Jing Zhou, Mingchao Zhou.

**Formal analysis:** Fang Liu, Alice Y. M. Jones, Raymond C. C. Tsang, Mingchao Zhou.

**Funding acquisition:** Fang Liu, Yulong Wang.

**Investigation:** Fang Liu, Raymond C. C. Tsang.

**Methodology:** Fang Liu, Alice Y. M. Jones.

**Project administration:** Fang Liu, Yulong Wang.

**Resources:** Fang Liu, Alice Y. M. Jones, Raymond C. C. Tsang, Yulong Wang.

**Supervision:** Fang Liu, Yulong Wang.

**Validation:** Fang Liu, Raymond C. C. Tsang, Yulong Wang.

**Writing – original draft:** Alice Y. M. Jones.

**Writing – review & editing:** Fang Liu, Alice Y. M. Jones, Raymond C. C. Tsang.

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
