## [Decision Letter · Decision Letter 0]

2 Jan 2020

PONE-D-19-24687

Noninvasive investigation of the cardiodynamic response to 6MWT in people with stroke using impedance cardiography

PLOS ONE

Dear Professor Jones,

Thank you for submitting your manuscript to PLOS ONE. After careful consideration, we feel that it has merit but does not fully meet PLOS ONE’s publication criteria as it currently stands. Therefore, we invite you to submit a revised version of the manuscript that addresses the points raised during the review process.

We would appreciate receiving your revised manuscript by Feb 16 2020 11:59PM. To enhance the reproducibility of your results, we recommend that if applicable you deposit your laboratory protocols in protocols.io, where a protocol can be assigned its own identifier (DOI) such that it can be cited independently in the future. For instructions see: http://journals.plos.org/plosone/s/submission-guidelines#loc-laboratory-protocols

We look forward to receiving your revised manuscript.

Kind regards,

Shane Patman, PhD

Academic Editor

PLOS ONE

Journal Requirements:

2. We note that Figures 1 and 2 includes an image of a [patient / participant / in the study]. 

Reviewers' comments:

Reviewer's Responses to Questions

**Comments to the Author**

1. Is the manuscript technically sound, and do the data support the conclusions?

Reviewer #1: Partly

Reviewer #2: Yes

2. Has the statistical analysis been performed appropriately and rigorously? 

Reviewer #1: No

Reviewer #2: Yes

3. Have the authors made all data underlying the findings in their manuscript fully available?

Reviewer #1: Yes

Reviewer #2: Yes

4. Is the manuscript presented in an intelligible fashion and written in standard English?

Reviewer #1: No

Reviewer #2: Yes

5. Review Comments to the Author

Reviewer #1: The subject of this paper is interesting and the attempt to use impedance cardiography in dynamic exercise tests is valuable. However, the scientific value of the concept of the study is not very high and methods have some important limitations.

1. Title (and whole article): The formula „patient with stroke” is for me misleading. It would be better „patients after stroke”, they were not with acute stroke.

2. Abstract: „better performed 6MWT”? was it always „better performed 6MWT”? From methods section I can conclude it was the second test, I suppose it was not better test for every subject. It should be explained.

3. Introduction – verse 67-69 – what is the purpose to put the sentence … „We hypothesized….” What it means in the context of the aims? Why did you hypothesize like that? How it is related with the paper aim?

4. Methods – in my opinion it is not enough to perform two 6MWT to prove that the method is repeatable. It was shown by previous researchers that in the second test distance is usually longer then in the first test.

5. Methods – inclusion/exclusion criteria (verse 82-92) – „duration of stroke”? – see comment 1.; „patients on beta blocker” – ever? While tested? ; „uncontrolled hypertension” – what cut-off, if all subjects presented normal BP? What was the mean value of BP before 6MWT?

6. Methods – were you really able to keep the temperature 23 C and humidity 60% (verse 110-111) at „hospital hallway”?

7. Methods – verse 162-164 – what variables were chosen for linear regression models, besides SV and CO? Where are the results of linear regression mentioned in verse 281-283 (with age and gender)?

8. Were there any withdrawals? The assumption for sample size was 20%.

9. Results – for summary description of study group statistics the word „mean” should by applied, i.e. „by MEAN 3,5 min…”

10. Patients functional/neurological state should be described in detail and commented if it could influence the capability to perform the 6MWT. It might influence the results and explain low max HR, what is discussed in verse 242-244.

11. NHISS and MRMI scores should be described in Methods.

12. Table 2 – the differences in HR between current study and most of other should be commented, some studies (ref. 29, 35) evaluated SV, CO, CI, it should be commented (what method, why your .

13. Table 2 – „EF change (%)” – it should be explained – is it value of EF change on units (%) or relative change in % of basal EF.

14. Table 3 is to complictated, some data should not be presented. The description

15. Discussion – verse 248-251 – repeated results.

16. Discussion verse 255-257 – the descirption of Tonelli study should be more correlated with previous text.

17. Discussion – verse 258-262 , related to Fig 2 – where are the results for 12 months?

18. What is „chronc stroke”? verse 280? What does suport the sentence „increase in CO was predominantly a consequence od SV increase …” verse 279-280

19. The abbrreviations should be introduced with first use and than used within further text.

20. Conclusions – it was not supported by the results that „SV contributed more when the time elapsed since stoke….” Verse 292. For SV it was 17%, for HR 18%.

21. In my opinions Figures 1 and 4 should be omitted.

Reviewer #2: Manuscript is interesting and valuable. However, I have some questions and comments.

Line 16-17 "whereas HR, CO and CI continued to rise for 90 sec before plateau." Statement is unclear.

Line 130 Are you sure you mean CO not CI? You are writing about "body surface area ". Besides, heart rate was measured based on and R-R interval that derived from the electrocardiograph (ECG) not CO. Please correct.

Line 136 -141 Please do not duplicate information. Figure with arrangement of electrodes is enough, without description in main text. Do you have a one picture with the device placed on the patient with the electrodes?

Table 1 How many patients have been diagnosed with coronary artery disease, HFpEF and other serious cardiac diseases? Were there any differences in subgroups?

Line 176-177 I do not agree with "SV reached a steady plateau at 30 sec into the 6MWT, while HR, CO and CI reached a plateau after 90 sec". This is a non-physiological mechanism. An increase in HR and CO is observed until the end of 6MWT. Figure 3 and Figure 5 prove this. The increase in these parameters is not large, but seems continuous and gradual. Of course, I agree that the increase in CO is mainly the result of an increase in HR and not SV, and SV quickly reached plateau. This is a valuable observation.

Line 132 What is the value of haemodynamic parameters obtained in 1 second intervals during ICG? Is the average of value of parameters of the number of heartbeat in such an interval or the current value every 1 second? Please consider if beat-to-beat recorded data would be more reliable, especially when HR> 120 / min. Have additional ventricular or supraventricular beats been observed or have they occurred in past medical history? Did you observe artifacts while recording ICG data?

Table 2 How did you count the Ejection Fraction at rest and at the end of 6MWT? Was echocardiography performed? Please complete the description of the methodology.

Fig 4. Figure 4 seems to be useless, because you did not assess blood pressure during 6MWT or systemic vascular resistance. Values blood pressure at rest, at the end of the effort, and possibly the time to return it to baseline will be enough.

Fig5. We observe a big discrepancy between CO and CI. Did the subgroups differ significantly between BSA and BMI?

Line 243-244 Do you have any data on the difference in activity between subgroups? Was rehabilitation in the subgroup > 12 months after the stroke performed? Were they more fit, physically active?

6. PLOS authors have the option to publish the peer review history of their article (what does this mean?). If published, this will include your full peer review and any attached files.

Reviewer #1: No

Reviewer #2: Yes: Małgorzata Kurpaska MD, PhD, Military Institute of Medicine, Warsaw, Poland

---

## [Author Response · Author response to Decision Letter 0]

5 Feb 2020

Responses to reviewers have been included in the 'Rebuttal letter' as well.

Reviewer #1: 

We are grateful for Reviewer 1’s valuable comments. Please see our response to each point raised. 

1. Title (and whole article): The formula „patient with stroke” is for me misleading. It would be better „patients after stroke”, they were not with acute stroke.

We have now modified the reference to our patients and changed ‘patients with stroke’ to patients after stroke. 

2. Abstract: better performed 6MWT”? was it always „better performed 6MWT”? From methods section I can conclude it was the second test, I suppose it was not better test for every subject. It should be explained.

All except 3 patients performed slightly better in the second 6MWT. We have used the data from the ‘better’ performed 6MWT for each patient. We have now further clarified this in the abstract (line 14-15). 

3. Introduction – verse 67-69 – what is the purpose to put the sentence … „We hypothesized….” What it means in the context of the aims? Why did you hypothesize like that? How it is related with the paper aim?

This sentence is now deleted

4. Methods – in my opinion it is not enough to perform two 6MWT to prove that the method is repeatable. It was shown by previous researchers that in the second test distance is usually longer then in the first test.

Despite a slight increase in the distance covered over the second 6MWT, there was no statistically significant difference in the distance covered. This suggests that the ‘practice effect’ of the two 6MWTs was only small. The high ICC values of ICG data recorded on consecutive days provides support for the reliability of ICG measurement of the cardiodynamic variables. We have explained this further in the discussion. Please refer to page 16, lines 285-288.

5. Methods – inclusion/exclusion criteria (verse 82-92) – „duration of stroke”? – see comment 1.; “patients on beta blocker” – ever? While tested? ; „uncontrolled hypertension” – what cut-off, if all subjects presented normal BP? What was the mean value of BP before 6MWT?

We have further clarified our inclusion and exclusion criteria. “Duration of stroke” is replaced with ‘time elapsed after stroke diagnosis’; “patients on beta blocker” is replaced with ‘patients prescribed with regular beta blockers or those required beta blocker at the time of the study’; “uncontrolled hypertension” is ‘as advised by the attending medical practitioner’. Please refer to page 6, line 88-93. BP values before the 6MWT are displayed in the supplementary data table: S2 Table. 

6. Methods – were you really able to keep the temperature 23 C and humidity 60% (verse 110-111) at „hospital hallway”?

The 6MWTs were conducted in an indoor corridor outside the laboratory. The temperature and humidity were controlled throughout the whole hospital. This is clarified on Page 7, line 110-113.

7. Methods – verse 162-164 – what variables were chosen for linear regression models, besides SV and CO? Where are the results of linear regression mentioned in verse 281-283 (with age and gender)?Were there any withdrawals? The assumption for sample size was 20%.

We followed the method of linear regression for similar comparisons, as published by Tonelli et al 2013*. We have initially included Age and gender as independent variables in the linear regression analysis, but as these have no statistically significant effect, these variables were removed from the linear regression model. SV and CO remain the main variables of interest for comparison. We have further explained this in our Result section (page 12, line 210-214). We have also included an extra table which details the characteristics of the two subject groups. Please refer to supplementary table S5 Table. 

Only two patients could not complete the second 6MWT. This is now reported at Page 9, line 166-167.

*Tonelli AR, Alkukhun L, Arelli V, Ramos J, Newman J, McCarthy K, et al. Value of impedance cardiography during 6-minute walk test in pulmonary hypertension. Clin Transl Sci. 2013;6(6): 474-480.

8. Results – for summary description of study group statistics the word „mean” should by applied, i.e. „by MEAN 3,5 min…”

The word ‘mean’ is now included in the sentence. Please refer to Page 10, line 178.

9. Patients functional/neurological state should be described in detail and commented if it could influence the capability to perform the 6MWT. It might influence the results and explain low max HR, what is discussed in verse 242-244.

More information on the mobility status of the participants is now included in Table 1. We have also included more details of the mobility status of our subjects in our discussion (page 14, line 243-250)

10. NHISS and MRMI scores should be described in Methods.

The NHISS and MRMI scores were retrieved from the patients’ bed notes. Please refer to page 7, line 105-106, under Procedure in Methods.

11. Table 2 – the differences in HR between current study and most of other should be commented, some studies (ref. 29, 35) evaluated SV, CO, CI, it should be commented (what method, why your .

Comparison with the two studies that evaluated haemodynamic variables during a 6MWT with Impedance Cardiograpy is now discussed in more detail. Please refer to Page 14, line 255-272.

12. Table 2 – „EF change (%)” – it should be explained – is it value of EF change on units (%) or relative change in % of basal EF.

Ejection Fraction was computed by the ICG software. This was not reported in other published studies, so we have now deleted this from Table 2. 

13. Table 3 is to complicated, some data should not be presented. The description

Presentation of Table 3 was modelled on Tonelli and Colleagues’ published work. We considered it a meaningful way of expressing the contribution of HR or SV to CO. May we know which data are considered unnecessary? 

14. Discussion – verse 248-251 – repeated results.

This sentence is now reworded, and ‘repeated’ results are deleted.

15. Discussion verse 255-257 – the description of Tonelli study should be more correlated with previous text.

The discussion has now been reworded, and comparison with Someya’s work included. 

16. Discussion – verse 258-262 , related to Fig 2 – where are the results for 12 months?

Figure 2 presents the pooled data of all 29 subjects. Sub-group analysis comparing changes in subjects with the time elapsed after stroke-diagnosis of shorter or longer than 12 months, are displayed in Figure 3. 

17. What is „chronic stroke”? verse 280? What does support the sentence „increase in CO was predominantly a consequence od SV increase …” verse 279-280

The terminology of chronic stroke is no longer used in the text. The paragraph explaining the relationship between CO, SV and HR is now reworded.

18. The abbreviations should be introduced with first use and then used within further text.

We have re-checked that abbreviations are all introduced the first time they appear in the text. 

19. Conclusions – it was not supported by the results that „SV contributed more when the time elapsed since stoke….” Verse 292. For SV it was 17%, for HR 18%.

The conclusion is now reworded.

20. In my opinions Figures 1 and 4 should be omitted.

Figures 1 and 4 are now deleted.

Responses to Reviewer# 2 

We are grateful for Reviewer 2’s valuable comments, please see our responses to questions raised:

1. Line 16-17 "whereas HR, CO and CI continued to rise for 90 sec before plateau." Statement is unclear.

This sentence is now reworded. Please refer to page 2, line 16-18

2. Line 130 Are you sure you mean CO not CI? You are writing about "body surface area ". Besides, heart rate was measured based on an R-R interval that derived from the electrocardiograph (ECG) not CO. Please correct.

This sentence is now reworded. Please see page 8, line 131-136

3. Line 136 -141 Please do not duplicate information. Figure with arrangement of electrodes is enough, without description in main text. Do you have one picture with the device placed on the patient with the electrodes?

Text description is now deleted as suggested.

4. Table 1 How many patients have been diagnosed with coronary artery disease, HFpEF and other serious cardiac diseases? Were there any differences in subgroups?

The patient history was retrieved from hospital notes. Only 5 patients were diagnosed with coronary artery disease and there was no record of heart failure as a diagnosis. We are of the view that a subgroup analysis of such a small number would not be meaningful.

5. Line 176-177 I do not agree with "SV reached a steady plateau at 30 sec into the 6MWT, while HR, CO and CI reached a plateau after 90 sec". This is a non-physiological mechanism. An increase in HR and CO is observed until the end of 6MWT. Figure 3 and Figure 5 prove this. The increase in these parameters is not large, but seems continuous and gradual. Of course, I agree that the increase in CO is mainly the result of an increase in HR and not SV, and SV quickly reached plateau. This is a valuable observation.

The HR, SV and CO indeed continue to rise till the end of the 6MWT, however, the rise was much steeper in the first 90 sec for HR and the first 30 sec for SV; thereafter then the rise became more gradual (S3 Table, Fig 2). We have reworded the description of this phenomenon. Please refer to page 10, line 176-178. 

6. Line 132 What is the value of haemodynamic parameters obtained in 1 second intervals during ICG? Is the average of value of parameters of the number of heartbeat in such an interval or the current value every 1 second? Please consider if beat-to-beat recorded data would be more reliable, especially when HR> 120 / min. Have additional ventricular or supraventricular beats been observed or have they occurred in past medical history? Did you observe artifacts while recording ICG data? 

The EnduroTM was used to capture cardiography signals. Signals were recorded at 1-second intervals. We did not observe any abnormal ventricular or supraventricular conduction abnormality.

7. Table 2 How did you count the Ejection Fraction at rest and at the end of 6MWT? Was echocardiography performed? Please complete the description of the methodology.

Ejection fraction was computed by the software. As this was not reported in other studies listed in Table 2, we have now deleted this variable from Table 2.

8. Fig 4. Figure 4 seems to be useless, because you did not assess blood pressure during 6MWT or systemic vascular resistance. Values blood pressure at rest, at the end of the effort, and possibly the time to return it to baseline will be enough.

Figure 4 is now deleted. Blood pressures before and immediately after 6MWT are displayed in S2 Table 

9. Fig5. We observe a big discrepancy between CO and CI. Did the subgroups differ significantly between BSA and BMI?

There were no statistical differences in BMI between the two groups. However computed BSA was higher in the group with time elapsed after stroke-diagnosis longer than 12 months. This may explain the higher CI in this group, but the difference in CI did not reach statistical significance. We have now included an additional table (S5 Table) to illustrate the differences in characteristics between the groups. 

10. Line 243-244 Do you have any data on the difference in activity between subgroups? Was rehabilitation in the subgroup > 12 months after the stroke performed? Were they more fit, physically active?

Unfortunately, we do not have detailed information on the activity of our patients. We do however know that the subjects with a longer time-elapsed after stroke diagnosis engaged in a longer duration of a walking program. We have included a more detailed discussion of this point. Please refer to page 14, line 243-251.

---

## [Decision Letter · Decision Letter 1]

17 Mar 2020

PONE-D-19-24687R1

Noninvasive investigation of the cardiodynamic response to 6MWT in people after stroke using impedance cardiography

PLOS ONE

Dear Professor Jones,

Thank you for submitting your manuscript to PLOS ONE. After careful consideration, we feel that it has merit but does not fully meet PLOS ONE’s publication criteria as it currently stands. Therefore, we invite you to submit a revised version of the manuscript that addresses the points raised during the review process.

I have been fortunate in being able to continue with the same two content expert peer reviewers from the original submission with this latest review cycle. Both reviewers have noted positive changes with this revised submission, however a few recommendations have arisen from this last peer review, as outlined below.

We would appreciate receiving your revised manuscript by May 01 2020 11:59PM. To enhance the reproducibility of your results, we recommend that if applicable you deposit your laboratory protocols in protocols.io, where a protocol can be assigned its own identifier (DOI) such that it can be cited independently in the future. For instructions see: http://journals.plos.org/plosone/s/submission-guidelines#loc-laboratory-protocols

We look forward to receiving your revised manuscript.

Kind regards,

Shane Patman, PhD

Academic Editor

PLOS ONE

Reviewers' comments:

Reviewer's Responses to Questions

**Comments to the Author**

1. If the authors have adequately addressed your comments raised in a previous round of review and you feel that this manuscript is now acceptable for publication, you may indicate that here to bypass the “Comments to the Author” section, enter your conflict of interest statement in the “Confidential to Editor” section, and submit your "Accept" recommendation.

Reviewer #1: All comments have been addressed

Reviewer #2: (No Response)

2. Is the manuscript technically sound, and do the data support the conclusions?

Reviewer #1: Yes

Reviewer #2: Partly

3. Has the statistical analysis been performed appropriately and rigorously? 

Reviewer #1: Yes

Reviewer #2: Yes

4. Have the authors made all data underlying the findings in their manuscript fully available?

Reviewer #1: Yes

Reviewer #2: Yes

5. Is the manuscript presented in an intelligible fashion and written in standard English?

Reviewer #1: Yes

Reviewer #2: Yes

6. Review Comments to the Author

Reviewer #1: Thank you for your work aimed to address my comments.

It is rather pilot study on a small sample size.

However, the conclusions are encouraging to perform further research in this area.

Reviewer #2: Thank you for answering my comments raised in a previous round of review. Unfortunately I have a lot of questions and remarks. I hope my comments will be useful and they will serve to improve the manuscript.

1. Line 45-47 The sentence: "The assessment... capacity." is not needed.

2. Line 61 Are you sure that references 15-18 are about exercise ICG?

3. Line 63 References 19-21 I think that more accurate study is for example: doi: 10.1007/s10554-019-01738-y than studies carried out in children.

4. Line 110-111 What is the significance of constant temperature and humidity for 6 MWT, please provide refferences.

5. Line 165-166 and line 169-170 repeated sentence "The distance covered in the second 6MWT was slightly greater than the first 6MWT"

6. Line 190-191 Do you know other studies (on other populations) describing the correlation of 6MWT with SV, HR, CO? Describe in the discussion.

7. Table 2 Are there studies with the use of 6MWT in case of diseases that are etiopathologically close to stroke? Hemodynamic studies during 6MWT or CPET with CAD, with AH, lack of training (sedentary lifestyle)? I understand that the model study was Tonelli's study but the PH pathogenesis is radically different from stroke, unless we are interested in the control group. Please comment on absolute values and parameter changes, are they larger / smaller? What can affect the observed hemodynamic profile in patients after stroke?

8. Line 203-204 The absolute values between the subgroups did not differ significantly, is not it? How can you explain the different percentage of linear regression? Were different trends in parameter changes observed between the subgroups? Which trend of changes is correct? Is any parameter trend a compensatory response? What could be the reason for the different results? Chronotropic failure? SV disability? How can you explain such a difference in the share of HR and SV in CO?

9. In the subgroup > 12 months after stroke, worse physical capacity was observed, but a greater relationship between CO and SV and HR. See DOI: 10.1080/10641963.2018.1523917

10. Line 223 - 228 should be put in the introduction rather than in the discussion.

11. According Ref. 39 "6MWT is not, by itself, an adequate measure of aerobic fitness early after stroke". Please explain the choice of 6MWT as a research method in a different way.

12. Line 230 -237 do not repeat information: The mean distance covered by our cohort of patients was 255m;... comparable to 274 m reported by Pradon" oraz "...that Pradon’s cohort achieved similar distance to our cohort...". Please edit these sentences.

13. Line 186-188 and line 237-239 Does it mean that 6MWT is performed according to the maximum capabilities? What could affect this result?

14. Line 239-242 What is the influence of 6MWT " with or without a walking stick, although some were unable to independently negotiate stair climbing". Please give the references. Was MRMI score result good or poor? Does it influence 6MWT, please give references.

15. Line 250-256 Please at first discuss your results and then comment this on other research. Do not describe other authors's researches.

16. Line 260-266 "Tonelli.... [29]" What does this information bring in to discussion on the hemodynamic response in stroke patients? I understand that you methodologically modeled the study on Tonelli's work, however this can be entered in the methodology. What is the relationship between PH and stroke, how to explain the similarity of hemodynamics?

17. Line 268 "While the changes in CO were similar". Were SV and HR changes also similar?

18. Line 274-278 How do you explain the shorter 6MWTD after a year? Did they exercise more or less? Are you suggesting that stroke patients in the study group had LVDD? In reply to question 4 was: "there was no record of heart failure as a diagnosis". What is the connection to LVDD?

19. Line 279- 280 Is this observation exceptional? See: doi: 10.1164/rccm.200203-166OC, doi: 10.1016/S0002-8703(03)00119-4,

20. Line 288-289 What research was the basis for suspecting a relationship between the age and sex with hemodynamic parameters? Please provide references.

7. PLOS authors have the option to publish the peer review history of their article (what does this mean?). If published, this will include your full peer review and any attached files.

Reviewer #1: No

Reviewer #2: No

---

## [Author Response · Author response to Decision Letter 1]

3 Apr 2020

Responses to Reviewer 2’s Comments

1. Line 45-47 The sentence: "The assessment... capacity." is not needed.

This sentence was not intended to be read in isolation but together with the fact that 6MWT is commonly used for assessment of aerobic capacity in people with stroke. Please see page 4, line 45 to 47. 

2. Line 61 Are you sure that references 15-18 are about exercise ICG?

References 15 to 18 are now replaced with a recent review article on clinical application of ICG [new reference 17]. Please kindly see page 5, line 61-63. 

3. Line 63 References 19-21 I think that more accurate study is for example: doi: 10.1007/s10554-019-01738-y than studies carried out in children.

We have now deleted the references to paediatric patients and replaced them with more recent publications, including the one suggested by the reviewer [Reference 18]. 

4. Line 110-111 What is the significance of constant temperature and humidity for 6 MWT, please provide references.

It is well known that variable temperature and humidity may influence exercise performance. Our subject cohort performed the sub-maximal exercise test in an environment where humidity and temperature were controlled. These criteria are not required by the ATS Guideline and so in order not to cause confusion, we have now removed the information on the temperature and humidity.

5. Line 165-166 and line 169-170 repeated sentence "The distance covered in the second 6MWT was slightly greater than the first 6MWT"

The first sentence refers to the ‘mean’ distance recorded and later refers to individual performance. To avoid the confusion, we have removed the second sentence. Please refer to line 166-169.

6. Line 190-191 Do you know other studies (on other populations) describing the correlation of 6MWT with SV, HR, CO? Describe in the discussion.

We have now added a paragraph describing correlation of 6MWT with SV, HR, and CO and compared our data with those reported in other populations. Please refer to page 15, line 250-260.

7. Table 2 Are there studies with the use of 6MWT in case of diseases that are etiopathologically close to stroke? Hemodynamic studies during 6MWT or CPET with CAD, with AH, lack of training (sedentary lifestyle)? I understand that the model study was Tonelli's study but the PH pathogenesis is radically different from stroke, unless we are interested in the control group. Please comment on absolute values and parameter changes, are they larger / smaller? What can affect the observed hemodynamic profile in patients after stroke?

Table 2 illustrates four studies [31,30,28,5] reporting heart rate changes associated with 6MWT. However SV, CO and CI data were not available in these studies. Our study is the first to report the SV, CO and CI response to a 6MWT in people after stroke. We have now highlighted that the ‘change’ in heart rate refers to an ‘increase’ in heart rate. Please refer to page 14, line 232.

8. Line 203-204 The absolute values between the subgroups did not differ significantly, is not it? How can you explain the different percentage of linear regression? Were different trends in parameter changes observed between the subgroups? Which trend of changes is correct? Is any parameter trend a compensatory response? What could be the reason for the different results?

We presume the reviewer is referring to lines 204-213 in the previous version. Linear regression analysis was a method used by Tonelli and colleagues [Reference 25] to determine whether the primary contribution to CO was HR or SV. We considered it meaningful to assess whether people with a longer post-stroke duration responded differently to exercise stress in the form of 6MWT. We have therefore adopted the same statistical method used by Tonelli and colleagues. 

9. In the subgroup > 12 months after stroke, worse physical capacity was observed, but a greater relationship between CO and SV and HR. See DOI:

10.1080/10641963.2018.1523917

 The reference suggested by the reviewer compared exercise capacity assessed via a 6MWT and cardiopulmonary exercise test with haemodynamic assessment via ICG in patients with arterial hypertension. The study concluded that ICG is a reliable method for assessing cardiovascular response to exercise. We are unsure how this relates to physical capacity in our subjects >12months after stroke. 

10. Line 223 - 228 should be put in the introduction rather than in the discussion.

We have now rewritten the first paragraph of our Discussion. Please refer to lines 221-228.

11. According Ref. 39 "6MWT is not, by itself, an adequate measure of aerobic fitness early after stroke". Please explain the choice of 6MWT as a research method in a different way.

Justification for the use of 6MWT in the assessment of cardiorespiratory fitness for people after stroke, is now included with an additional reference. Please see lines 221-228.

12. Line 230 -237 do not repeat information: The mean distance covered by our cohort of patients was 255m;... comparable to 274 m reported by Pradon" oraz "...that Pradon’s cohort achieved similar distance to our cohort...". Please edit these sentences.

We were confused as to whether the reviewer prefers us using the absolute values to illustrate our discussion point or a ‘similar’ distance statement was sufficient. The second sentence, which appears to reiterate that a similar distance was covered in both studies, is in fact necessary to highlight that ‘despite the similar distance’, the reported rate of perceived exertion was very different. Please refer to line 230-236.

13. Line 186-188 and line 237-239 Does it mean that 6MWT is performed according to the maximum capabilities? What could affect this result?

It is expected that a 6MWT is performed at the subject’s maximum capacity. While the VO2peak achieved at the end of a 6MWT is very similar to that obtained from a cycle graded exercise test (for example current reference 35), a graded cardiopulmonary exercise test remains the gold standard for assessment of maximum capacity. We followed the ATS guideline and delivered standard instructions to our subjects to attain their maximum performance. The results (% MHR at the end of 6MWT) are comparable to data reported by others, so we do not anticipate a problem with this methodology. The aim of our study was to illustrate that ICG can be used to measure haemodynamic parameters during a 6MWT and easily applied in a clinical setting. Construct validity of 6MWT performance as a measure of functional walking capacity in people with acute, subacute and chronic stroke has been well established [Reference36], and is beyond the scope of this current study. 

14. Line 239-242 What is the influence of 6MWT " with or without a walking stick, although some were unable to independently negotiate stair climbing". Please give the references. Was MRMI score result good or poor? Does it influence 6MWT, please give references.

Our subjects were able to mobilize independently although a few required a walking cane. The number of those who required a walking cane is small and therefore subgroup analysis is not appropriate. ATS guidelines only require a record of whether the subject required a walking aid or not. Table 1 illustrates the high MRMI score in our patients. This is further explained in the discussion (please refer to page 14, line 240-241). The MRMI score reflects the mobility level of our cohort and correlates well with physical function variables such as range of movement and sensory function (doi:10.1589/jpts.28.2389). That being said, it is not a measurement tool for aerobic capacity. Detailed investigation of the relationship between MRMI and 6MWD is beyond the scope of our study.

15. Line 250-256 Please at first discuss your results and then comment this on other research. Do not describe other authors's researches.

We appreciate this may be true in most circumstances, however in our view, the concept would be more difficult for the readership to grasp if we commenced with data from a population of stroke survivors. We consider it is easier to first explain the scenario in a normal healthy population in various age groups. 

16. Line 260-266 "Tonelli.... [29]" What does this information bring in to discussion on the hemodynamic response in stroke patients? I understand that you methodologically modeled the study on Tonelli's work, however this can be entered in the methodology. What is the relationship between PH and stroke, how to explain the similarity of hemodynamics?

We have now deleted the paragraph reiterating the linear regression method employed by Tonelli. 

17. Line 268 "While the changes in CO were similar". Were SV and HR changes also similar?

Yes, all changes were similar. Please refer to supplementary table 5. ( S5 Table 5).

18. Line 274-278 How do you explain the shorter 6MWTD after a year? Did they exercise more or less? Are you suggesting that stroke patients in the study group had LVDD? In reply to question 4 was: "there was no record of heart failure as a diagnosis". What is the connection to LVDD?

The shorter 6MWD shown in Table 3 does not refer to a shorter 6MWD ‘after a year’. Please refer to Table 1, which shows that the duration of post stroke time in our subjects ranged from 1 month to 60 months. These sentences only postulate that diastolic function in patients after stroke may be influenced by exercise or activities. We did not at all suggest that our patients have left ventricular diastolic dysfunction. We have reworded the respective sentence to reflect our speculation. Please see page 16, line 281-282. 

19. Line 279- 280 Is this observation exceptional? See: doi: 10.1164/rccm.200203-166OC, doi: 10.1016/S0002-8703(03)00119-4,

The learning effect with 6MWT is well documented in people with respiratory disease, however in people after stroke, reliability was shown to be excellent with one 6MWT trial and a single 6MWT is recommended in people after stroke (Reference 36). This is now highlighted in the Discussion, page 16, line 284-286.

20. Line 288-289 What research was the basis for suspecting a relationship between the age and sex with hemodynamic parameters? Please provide references.

We have now included two references [39,40] to support our view that age and gender influence cardiovascular function parameters. Please see page 17, line 295-296.

---

## [Decision Letter · Decision Letter 2]

17 Apr 2020

PONE-D-19-24687R2

Noninvasive investigation of the cardiodynamic response to 6MWT in people after stroke using impedance cardiography

PLOS ONE

Dear Professor Jones,

Thank you for submitting your manuscript to PLOS ONE. After careful consideration, we feel that it has merit but does not fully meet PLOS ONE’s publication criteria as it currently stands. Therefore, we invite you to submit a revised version of the manuscript that addresses the points raised during the review process.

We would appreciate receiving your revised manuscript by Jun 01 2020 11:59PM. To enhance the reproducibility of your results, we recommend that if applicable you deposit your laboratory protocols in protocols.io, where a protocol can be assigned its own identifier (DOI) such that it can be cited independently in the future. For instructions see: http://journals.plos.org/plosone/s/submission-guidelines#loc-laboratory-protocols

We look forward to receiving your revised manuscript.

Kind regards,

Shane Patman, PhD

Academic Editor

PLOS ONE

Additional Editor Comments (if provided):

As academic editor I have been fortunate, even during this uncertain worldly times, to have secured consistency with reviewers for this R2 submission. Reviewer 2 remains with some significant concerns, as outlined below, which require further opportunity for rebuttal commentary from authors at this stage

Reviewers' comments:

Reviewer's Responses to Questions

**Comments to the Author**

1. If the authors have adequately addressed your comments raised in a previous round of review and you feel that this manuscript is now acceptable for publication, you may indicate that here to bypass the “Comments to the Author” section, enter your conflict of interest statement in the “Confidential to Editor” section, and submit your "Accept" recommendation.

Reviewer #1: All comments have been addressed

Reviewer #2: (No Response)

2. Is the manuscript technically sound, and do the data support the conclusions?

Reviewer #1: Yes

Reviewer #2: Partly

3. Has the statistical analysis been performed appropriately and rigorously? 

Reviewer #1: Yes

Reviewer #2: Yes

4. Have the authors made all data underlying the findings in their manuscript fully available?

Reviewer #1: Yes

Reviewer #2: Yes

5. Is the manuscript presented in an intelligible fashion and written in standard English?

Reviewer #1: Yes

Reviewer #2: Yes

6. Review Comments to the Author

Reviewer #1: I have no additional comments. All previous comments were addressed well. The manuscript is now correct .

Reviewer #2: Major Comments:

The manuscript is valuable in terms of methodology and results. Unfortunately, I have the impression that you are completely unable to explain whether the observed hemodynamic reaction is correct or not, whether it have been observed in other groups (not in PH!) already and what is clinical significance of your observations. I have no doubts about the 6MWT methodology. The intention of my questions 6 - 9 and 13 and 18 was a broader view on the obtained results in order to supplement and improve the discussion on their basis. Compare the obtained hemodynamic results to the results obtained in other, more pathophysiologically related diseases to stroke such as CAD or AH.

Ad Q 9:

Do you know the entire manuscript? In this manuscript DOI 10.1080/10641963.2018.1523917:

Stronger correlations were observed between absolute value of VO2 and HR and VO2 and CO at peak exercise and changes in all of the evaluated parameters in subgroup of patient and reduced exercise capacity than with normal peak VO2 (>80% pred.)

I think this information is worth of consideration in the discussion.

Line 273-280

What can be the reason for the different contribution to CO by HR and SV observed by Comparing subgroups of patients: "time elapsed after stroke less than 1 year" and "time elapsed after stroke over 1 year"? Information on left ventricular dysfunction is insufficient in my opinion.

Minor Comments:

Line 223-230 Good justification for the choice of method. It is recommended to move this fragment to the introduction as before. Please discuss the results in the discussion.

Line 242 Please complete the discussion with the sentence: The MRMI score reflects the mobility level of our cohort and correlates well with physical function variables such as range of movement and sensory.

Line 281 Please replace "but" with "and".

7. PLOS authors have the option to publish the peer review history of their article (what does this mean?). If published, this will include your full peer review and any attached files.

Reviewer #1: No

Reviewer #2: No

---

## [Author Response · Author response to Decision Letter 2]

20 Apr 2020

Responses to Reviewer 2’s comments 

Major Comments: 

The manuscript is valuable in terms of methodology and results. Unfortunately, I have the impression that you are completely unable to explain whether the observed hemodynamic reaction is correct or not, whether it have been observed in other groups (not in PH!) already and what is clinical significance of your observations. I have no doubts about the 6MWT methodology. The intention of my questions 6 - 9 and 13 and 18 was a broader view on the obtained results in order to supplement and improve the discussion on their basis. Compare the obtained hemodynamic results to the results obtained in other, more pathophysiologically related diseases to stroke such as CAD or AH.

We are disappointed that Reviewer 2 has the misconception that we are ‘completely unable’ to explain whether the observed hemodynamic reaction is correct or not and ‘whether it have been observed in other group (not in PH!)’. 

We believe that Reviewer 2 appears to have failed to give appropriate weight to the primary aims of our study. 

Our paper aims to report the cardiodynamic parameters recorded by Impedance Cardiography (ICG) DURING a 6MWT in a cohort of people after stroke. Our paper is the first to report SV, CO, CI during a 6MWT. The manuscript describes cardiodynamic parameters recorded by ICG during a 6MWT. There are limited publications from which to draw a reference and none for direct comparison, but our focus is cardiac parameters recorded DURING a 6MWT.

In a nutshell, our study describes the rationale for obtaining cardiodynamic data during a common clinical aerobic capacity test for people after stroke; explains how ICG was applied; reports the data recorded; and compares our data (recorded by ICG DURING a 6MWT) with reported literature on cardiodynamic variables obtained by ICG DURING a 6MWT in other patient cohorts. 

The focus of our paper was on the potential role for ICG in the examination of cardiodynamic function in stroke patients. Tonelli et al. 2013 and Someya et al. 2015 were two excellent papers that describe clearly how to determine whether the main contributor to CO is SV or HR. Therefore, although these papers do not describe data from people with stroke, their work explains the analysis of cardiodynamic responses acquired by ICG during a 6MWT, highlighting the potential for ICG in further investigation during stroke rehabilitation; we believe their work is worthy of consideration. 

In short, the focus of this article is not the cardiodynamics and aerobic capacity of people with stroke, but the measuring capability of ICG during a 6MWT in people with stroke. 

Ad Q 9:

Do you know the entire manuscript? In this manuscript DOI 10.1080/10641963.2018.1523917:

Stronger correlations were observed between absolute value of VO2 and HR and VO2 and CO at peak exercise and changes in all of the evaluated parameters in subgroup of patient and reduced exercise capacity than with normal peak VO2 (>80% pred.)

I think this information is worth of consideration in the discussion.

Yes, we have considered and understand this article. This article described the cardiodynamic parameters obtained by ICG during a cardiopulmonary exercise test (CPET). The subjects also were subjected to a 6MWT. The correlation between the 6-minute walk distance (6MWD) achieved and the peak VO2, HR, SV and CO obtained during the CPET was investigated. This paper did NOT access cardiodynamic data DURING a 6MWT and therefore is irrelevant to the focus of our paper. This article forms the basis of the second phase of our study, which compares cardiodynamic data obtained by ICG during CPET and ICG data obtained during 6MWT. 

Line 273-280

What can be the reason for the different contribution to CO by HR and SV observed by Comparing subgroups of patients: "time elapsed after stroke less than 1 year" and "time elapsed after stroke over 1 year"? Information on left ventricular dysfunction is insufficient in my opinion.

In our discussion we suggest that the lesser contribution of SV to CO might be associated with diastolic dysfunction. The contribution of SV to CO appeared to be higher in the patients in our cohort with the opportuity for a longer period of rehabiliation training. This does nothing more than invite further investigation of the contributions of SV and HR to CO in response to exercise training. ICG is a convenient modality to determine this information. It is not the aim of our paper to discuss left ventricular dysfunction in our subject cohort. 

Minor Comments: 

Line 223-230 Good justification for the choice of method. It is recommended to move this fragment to the introduction as before. Please discuss the results in the discussion.

Line 242 Please complete the discussion with the sentence: The MRMI score reflects the mobility level of our cohort and correlates well with physical function variables such as range of movement and sensory.

Line 281 Please replace "but" with "and".

The above has been addressed accordingly.

---

## [Decision Letter · Decision Letter 3]

28 Apr 2020

Noninvasive investigation of the cardiodynamic response to 6MWT in people after stroke using impedance cardiography

PONE-D-19-24687R3

Dear Dr. Jones,

We are pleased to inform you that your manuscript has been judged scientifically suitable for publication and will be formally accepted for publication once it complies with all outstanding technical requirements.

With kind regards,

Shane Patman, PhD

Academic Editor

PLOS ONE

Additional Editor Comments (optional):

Reviewers' comments:

Reviewer's Responses to Questions

**Comments to the Author**

1. If the authors have adequately addressed your comments raised in a previous round of review and you feel that this manuscript is now acceptable for publication, you may indicate that here to bypass the “Comments to the Author” section, enter your conflict of interest statement in the “Confidential to Editor” section, and submit your "Accept" recommendation.

Reviewer #2: All comments have been addressed

2. Is the manuscript technically sound, and do the data support the conclusions?

Reviewer #2: (No Response)

3. Has the statistical analysis been performed appropriately and rigorously? 

Reviewer #2: (No Response)

4. Have the authors made all data underlying the findings in their manuscript fully available?

Reviewer #2: (No Response)

5. Is the manuscript presented in an intelligible fashion and written in standard English?

Reviewer #2: (No Response)

6. Review Comments to the Author

Reviewer #2: (No Response)

7. PLOS authors have the option to publish the peer review history of their article (what does this mean?). If published, this will include your full peer review and any attached files.

Reviewer #2: No

---

## [Editor Report · Acceptance letter]

8 May 2020

PONE-D-19-24687R3 

Noninvasive investigation of the cardiodynamic response to 6MWT in people after stroke using impedance cardiography 

Dear Dr. Jones:

I am pleased to inform you that your manuscript has been deemed suitable for publication in PLOS ONE. Congratulations! Your manuscript is now with our production department. 

With kind regards,

on behalf of

Assoc Prof Shane Patman 

Academic Editor

PLOS ONE